# Boosting Electrocatalytic Oxidation of Formic Acid on Ir(IV)-Doped PdAg Alloy Nanodendrites with Sub-5 nm Branches

**DOI:** 10.3390/molecules28093670

**Published:** 2023-04-23

**Authors:** Gongguo Zhang, Yingying Wang, Yanyun Ma, Haifeng Zhang, Yiqun Zheng

**Affiliations:** 1School of Chemistry, Chemical Engineering, and Materials, Jining University, Qufu 273155, China; 2Health Management Department, Shandong Vocational College of Light Industry, Zibo 255300, China; 3Institute of Functional Nano & Soft Materials (FUNSOM), Jiangsu Key Laboratory of Advanced Negative Carbon Technologies, Soochow University, Suzhou 215123, China

**Keywords:** electrocatalysts, nanocrystals, formic acid oxidation, doping, iridium

## Abstract

The formic acid oxidation reaction (FAOR) represents an important class of small organic molecule oxidation and is central to the practical application of fuel cells. In this study, we report the fabrication of Ir(IV)-doped PdAg alloy nanodendrites with sub-5 nm branches via stepwise synthesis in which the precursors of Pd and Ag were co-reduced, followed by the addition of IrCl_3_ to conduct an in situ galvanic replacement reaction. When serving as the electrocatalyst for the FAOR in an acidic medium, Ir(IV) doping unambiguously enhanced the activity of PdAg alloy nanodendrites and improved the reaction kinetics and long-term stability. In particular, the carbon-supported PdAgIr nanodendrites exhibited a prominent mass activity with a value of 1.09 A mg_Pd_^−1^, which is almost 2.0 times and 2.7 times that of their PdAg and Pd counterparts, and far superior to that of commercial Pt/C. As confirmed by the means of the DFT simulations, this improved electrocatalytic performance stems from the reduced overall barrier in the oxidation of formic acid into CO_2_ during the FAOR and successful d-band tuning, together with the stabilization of Pd atoms. The current study opens a new avenue for engineering Pd-based trimetallic nanocrystals with versatile control over the morphology and composition, shedding light on the design of advanced fuel cell electrocatalysts.

## 1. Introduction

The formic acid oxidation reaction (FAOR) is of significant importance in the fundamental research of small organic molecule electro-oxidation and the practical application of fuel cells [1,2]. It is well accepted that achieving a satisfactory performance with the direct use of commercial Pd/C as an electrocatalyst remains difficult, as it experiences CO poisoning and sharp current density decay after long-term cycling [3]. To this end, the precise engineering of Pd-based electrocatalysts is essential, and alloying Pd with a secondary metal represents a feasible option [4,5,6,7]. This is because, according to the calculations by Nørskov et al., the d-band center of Pd can be manipulated to obviously up-shift or down-shift upon alloying, depending on the change in the surface tensile strain [8]. The change in electronic properties could cause the variation in surface affinity between Pd atoms and key reaction intermediates, which then affects the electrocatalytic pathways [9,10].

Until now, tremendous research efforts have been devoted to synthesizing Pd-based nanoalloys (i.e., PdPt [11,12], PdSn [13], PdCo [14,15], PdCu [16], PdCuAu [17], et al. [18]) and evaluating their performance in the electrocatalytic FAOR [19]. Taking PdAg as an example [20,21], Tang and coworkers reported the fabrication of polyhedral PdAg alloy nanoparticles, which exhibited higher electrocatalytic activity and stability than commercially available Pd black [22]. Later, a facile synthesis for channel-rich ultrathin palladium–silver nanosheets was developed, which showed a mass activity of 987 mA mg^−1^ [23]. In addition to the electrochemical performance advantages, the doping of Ag also shows great merits in reducing the overall cost of Pd-based electrocatalysts.

To further optimize Pd-based electrocatalysts, more elements are expected to be included to modify the electronic structure and chemical environment of Pd atoms [24,25,26,27,28]. Generally, the FAOR contains several sets of important elementary steps and the exact catalytic mechanisms are still unclear (Appendix A). According to theoretical analyses, electrocatalysts for the FAOR should have lower onset potentials for direct oxidation via the formate pathway, higher onset potentials for forming COOH* to avoid CO* formation, and lower onset potentials for indirect oxidation via the carboxyl pathway to facilitate removing CO* [29]. Based on these mechanisms, it was theoretically deduced that alloying Pd using reactive metals, such as Ir, Rh, and Ru, could meet two out of the three conditions. For example, alloying could help stabilize HCOO* relative to the pure host metals and, thus, removing HCOO* becomes the potential-determining step.

Among recent studies, we have witnessed a rapid development in the rational design of high-performance electrocatalysts based on the Ir-doping strategy [30]. For example, Guo and coworkers reported an iridium–tungsten alloy with a nanodendritic structure as a new class of high-performance and pH-universal bifunctional electrocatalysts for hydrogen and oxygen evolution catalysis, which presents a hydrogen generation rate ∼2 times higher than that of the commercial Pt/C catalyst in both acid and alkaline media [31]. Yu and coworkers developed a facile approach to control the composition of IrPt alloy nanoparticles without affecting the size using an antisolvent crystallization-based method, based on the separation of particle growth and the precursor reduction steps, which exhibited satisfactory electrocatalytic performances in both the hydrogen evolution reaction and oxygen reduction reaction [32]. Henkelman and coworkers reported a rational synthetic strategy for the preparation of sub-10 nm Rh-Ir nanoparticles, which are highly efficient oxygen evolution reaction catalysts under acidic conditions, based on microwave-assisted synthesis [33]. All these successful demonstrations illustrate the feasibility of involving Ir as a promising catalytic element in constructing efficient electrocatalysts.

Herein, we report the water-based synthetic strategy of PdAg nanodendrites (NDs) with sub-5 nm branches, together with the Ir(IV) doping strategy realized via the galvanic replacement reaction (GRR) under ambient conditions. The variation in structure and compositions induced by Ir(IV) doping was systematically tracked and analyzed using XRD and XPS. The effect of the capping agent on the product morphology was investigated by conducting control experiments using different quaternary ammonium surfactants. Both electrochemical measurements and theoretical simulations confirmed the positive role of doping in enhancing the specific activity, reaction kinetics, and long-term durability of Pd-based electrocatalysts.

## 2. Results and Discussion

### 2.1. Synthesis of PdAgIr NDs

The typical synthesis of PdAg NDs included the reduction of Pd and Ag precursors in the presence of octadecylammonium chloride (OTAC) at 60 °C. The resulting products showed the morphology of spherical assemblies constructed of multiple nanobranches (Figure 1a and Appendix A). Abundant pores and channels could be observed in the particle body (Figure 1b,e). The diameter of the branch was measured to be less than 5 nm (Figure 1c). The limited size should be beneficial for the atomic exposure on the particle surface and, thus, the optimized atomic utilizations in electrocatalysis. The lattice fringe was measured to be 0.19 nm, corresponding to the (111) crystal plane of the PdAg alloy (see the inset of Figure 1c). Selected-area electron diffraction (SAED) pattern of an individual PdAg ND confirmed its polycrystalline structure (Figure 1d). Both elemental Pd and Ag were homogeneously distributed over the whole particle, as evidenced by the EDX-STEM mapping images and corresponding line-scan profile. The atomic ratio of Pd:Ag was determined by EDS to be 58.5:41.5.

After Ir doping was realized by the galvanic replacement reaction, the resulting products maintained their original morphology (Appendix A), but the diameter of the nanobranch increased slightly to 4.3 nm (Figure 2a) and the lattice fringe increased to 0.21 nm (Figure 2b), which was likely caused by Ir doping. The polycrystalline nature was also retained, as shown in the SAED pattern (Figure 2c). The HAADF-STEM image showed that noticeable distances were maintained between adjacent nanobranches (Figure 2d). EDX-STEM mapping images (Figure 2e–h) and the corresponding line-scan profile (Figure 2i) showed that Ir doping did not change the elemental distribution of either Pd or Ag, and the elemental Ir was also homogeneously distributed over the whole particle. The atomic ratio of Pd:Ag:Ir was determined to be 54.4:35.9:9.7. The decreased atomic percentage of both Pd and Ag indicated that the IrCl_3_ reacted with both elemental Pd and Ag during the galvanic replacement reaction.

### 2.2. XRD and XPS Analyses

XRD patterns were collected to show the change in the crystal structure caused by the GRR treatment. As shown in Figure 3, both PdAg NDs and PdAgIr NDs showed typical diffraction patterns of face-centered cubic (*fcc*) metals. Their diffraction peaks were located between the two peaks documented in the JCPDS database, which could be indexed to elemental Ag and Pd/Ir, respectively, suggesting the possible formation of an alloy phase. The detailed peak positions are summarized in Appendix A. Overall, by comparing the diffraction peak positions of PdAg NDs before and after the GRR treatment, we found that the GRR caused each of the diffraction peaks to shift, with an increase of 0.2~0.3 degrees, indicating the change in lattice parameters. This increase in two-theta values matched well with the results and trends predicted by theoretical simulations using Vegard’s Law. The crystalline sizes of PdAg NDs before and after the GRR treatment as determined via the Scherrer Equation are summarized in Appendix A. The difference between the branch size and the overall particle size as statistically measured in the corresponding electroscope image was likely due to the anisotropic dendritic shape.

We also studied the electronic states by conducting an XPS analyses. Typical doublets of Ag and Pd in zero-valent states were dominant in the corresponding Ag 3d and Pd 3d spectra, respectively (Figure 4a–d). After Ir doping, the slight shift in binding energy (B.E.) peaks for both elements could be attributed to the change in the electronic structure. Moreover, the fitted XPS peaks suggested the partial presence of Pd and Ag at a higher oxidation state, i.e., possibly PdO and Ag_2_O, respectively. The Ir 4f spectrum showed that the two B.E. peaks indexed to Ir(IV), suggesting the Ir species were present on the particle surface in oxidation states, e.g., IrO_2_ (Figure 4e). This observation was further validated by the corresponding EDS results, where the O Kα line located at 0.525 keV could be noted in the spectrum (Appendix A). The detailed B.E. peak positions and deconvolution parameters are summarized in Appendix A. To determine the d-band center, the valence band (VB) XPS spectra were collected. As shown in Figure 4f, the d-band centers of PdAg NDs and PdAgIr NDs were located at 5.04 eV and 5.30 eV, respectively. This showed that the overall engineering of the d-band orbitals of PdAg was successfully achieved by Ir doping.

### 2.3. Formation Mechanism of Dendritic Morphology

To elucidate the formation mechanism of a dendritic morphology, several control experiments were conducted. For example, a set of quaternary ammonium surfactants were chosen to replace OTAC at the same concentration, such as hexadecyltrimethylammonium chloride (CTAC), docosyltrimethylammonium chloride (DCTAC), dodecyltrimethylammonium chloride (DDTAC), benzethonium chloride (BC), octadecyltrimethylammonium bromide (OTAB), and hexadecyltrimethylammonium bromide (CTAB). As shown in Appendix A, the change in capping agent causes both variations in the size and morphology. Moreover, the use of capping agents, such as DDTAC and CTAB, resulted in a poor uniformity for final products. It showed that both the alkyl chain length and structure, and the anion of the quaternary ammonium surfactant played a crucial role in determining the size/shape of the final products. Thus, the rational screening of the capping agent with appropriate molecular structures is of great importance.

### 2.4. FAOR Measurements

The as-prepared GRR-treated PdAg NDs were purified and loaded on carbon black to work as electrocatalysts, denoted as PdAgIr NDs/C. Their electrochemically active specific surface areas (ECSAs) were measured by collecting cyclic voltammetry (CV) curves in 0.1 M HClO_4_ solution and calculated based on the integrated charge associated with the cathodic peak in the range of 0.6~1.0 V vs. RHE. The FAOR tests were conducted in 0.1 M HClO_4_ solution containing 0.5 M FA solution, where the CV curves were collected at a scan rate of 50 mV s^−1^. The long-term stability of these electrocatalysts were evaluated using chronoamperic (CA) measurements, where the *i-t* curves were recorded for 3000 s at 0.65 V vs. RHE. For comparison, the as-mentioned Pd NDs and PdAg NDs were purified and loaded on carbon to serve as the control group, while the commercial Pt/C electrocatalyst (Model No. HiSPEC4000, Johnson Matthey) was employed as the benchmark group.

As shown in Figure 5a, the ECSAs of the Pd NDs/C, PdAg NDs/C, and PdAgIr NDs/C electrocatalysts were 23.9 m^2^ g_Pd_^−1^, 43.2 m^2^ g_Pd_^−1^, and 52.7 m^2^ g_Pd_^−1^, respectively. The difference in the ECSA value could be attributed to the difference in porosity of the dendritic layer and the inter-distance between branches, which was improved by Ir doping. With the presence of FA, all CV curves were drastically changed due to FAOR electrocatalysis, where an intense and broad anodic wave was noticed in the forward scan. The mass activities for the FAOR of the Pd NDs/C, PdAg NDs/C, and PdAgIr NDs/C electrocatalysts were 396.9 mA mg_Pd_^−1^, 551.4 mA mg_Pd_^−1^, and 1086 mA mg _Pd_^−1^, respectively (Figure 5b). Specific activities were also calculated by normalizing the corresponding ECSA values (Appendix A), where the Pd NDs/C, PdAg NDs/C, and PdAgIr NDs/C electrocatalysts showed the specific activity of 16.6 A m^−2^, 12.8 A m^−2^, and 20.6 A m^−2^, respectively. Both results suggested that Ir doping towards the PdAg nanoalloys contributed to an enhanced FAOR activity. The relatively small onset potential of PdAgIr NDs/C suggested that the FAOR could be easier to initiate on its surface, as compared to PdAg NDs/C.

To elucidate the reaction kinetics, as shown in Figure 5c, CV curves in HClO_4_ + FA were also collected under different scan rates and the linear relationship between the square root of the scan rate (v^1/2^) (10–250 mV s^−1^) and the forward peak current density plotted according to the Randles–Sevcik equation. The PdAgIr NDs/C electrocatalysts showed an enhancement in electrocatalytic kinetics as concluded from a larger slope value as compared to the PdAg NDs/C and Pd NDs/C.

Figure 5d shows the CA measurements of the two electrocatalysts, where the PdAgIr NDs/C electrocatalyst exhibited a higher mass activity after cycling than those of the PdAg NDs/C and Pd NDs/C. In addition, compared to Pt/C, the value of the ECSA (Appendix A) and mass activity (Appendix A) observed on the PdAgIr NDs/C was nearly 4-fold and 12-fold that of the commercial Pt/C (i.e., ECSA(Pt/C) = 12.1 m^2^ g_Pt_^−1^, mass activity (Pt/C) = 91.7 mA mg_Pt_^−1^), respectively, together with the improved reaction kinetics (Appendix A) and cycling stability (Appendix A), showing their potential practical use as electrocatalysts for fuel cell devices. The as-mentioned data are summarized in Appendix A. Additionally, a comprehensive analysis of the recent literature further illustrated that the current PdAgIr electrocatalyst exhibited a relatively competitive FAOR mass activity (Appendix A).

### 2.5. DFT Simulations

The electrocatalytic activities of the PdAg NDs/C and PdAgIr NDs/C for the FAOR were then evaluated using DFT simulations. Two electrocatalysts, namely, the AgPd(111) and IrAgPd(111) catalysts, were modeled to probe the effect of Ir doping on electrocatalytic activity. For each catalyst, the overall barriers of the direct oxidation pathway and CO poisoning pathway were compared. The structures of the intermediates and reaction energy profiles for the FAOR are shown in Figure 6. The reactions were initiated from the most stable configurations of HCOOH (Figure 6a,b). For the direct oxidation pathway, the initial activation of HCOOH was very likely to have occurred at the O-H bond. We noticed that the overall barrier was 1.35 eV on AgPd(111) and 0.90 eV on IrAgPd(111) (Figure 6c), verifying that IrAgPd(111) had a better catalytic activity for the FAOR. For the CO poisoning pathway, CO would be produced through carboxyl, which agreed well with the previous theoretical and experimental results [29,34]. As showed in Figure 6d, AgPd(111) exhibited a larger overall barrier than IrAgPd(111) (2.04 eV vs. 1.95 eV), indicating a lower possibility for the CO poisoning effect on AgPd(111).

### 2.6. Mechanism for FAOR Activity Enhancement

Taking these data into consideration, we believed that the enhanced FAOR activities of Ir(IV)-doped PdAg NDs can mainly be ascribed to the geometric and electronic effects. For the electronic effect, by doping Ir(IV) with PdAg, the d-band center shifts away from the Fermi level (Figure 7), and the bond strength between the CO-based adsorbate and the Pd surface weakens due to the activated antibonding for the Pd-O. Consequently, Pd-O bond scission requires less energy and, thus, HCOO_ads_ decomposition is easier. For the geometric effect, the unique 3D dendritic structure with abundant pores and channels provided an enlarged surface area and abundant uncoordinated sites for the FAOR, which was also beneficial to avoid the aggregation of nanocatalysts during cycling [35].

In addition, it is well accepted that the coalescence of adjacent particles/crystalline boundaries is the key issue that causes the loss of active sites and, thus, the deactivation of the electrocatalysts during long-term cycling [36,37]. To solve this problem, effective strategies should be applied to inhibit surface atomic diffusion. In the current case, we hypothesized that the introduced Ir atoms would dampen surface atomic diffusion. To demonstrate this, DFT simulations of the vacancy formation energy (ΔE_v_) of Pd atoms were performed, where a larger ΔE_v_ suggested a stronger metal–metal interaction and weaker diffusion ability of the metal atom. As shown in Figure 8, four different models, including Pd, PdAg, PdIr, and PdAgIr, were constructed to investigate the change in the ΔE_v_ of the Pd atoms on the (111) facets after Ag/Ir doping, where two typical sites were selected for calculations. Compared with that of the Pd {111} slab (i.e., 0.71 eV/0.76 eV), the ΔE_v_ of Pd atoms in the PdAg {111} slab was lower (i.e., 0.65 eV/0.57 eV), but the ΔE_v_ of Pd atoms in the PdIr {111} slab was higher (i.e., 1.31 eV/1.32 eV). Moreover, the ΔE_v_ of Pd atoms in the PdAgIr {111} slab (i.e., 0.78 eV/0.70 eV) was also higher than that of both Pd and PdAg. These results suggested that the doping of Ir was crucial to stabilize the surface Pd atoms and trade off the negative effect brought about by Ag doping, which illustrated the structural advantages of trimetallic electrocatalysts as compared to binary ones.

## 3. Materials and Methods

**Materials.** All the chemicals were obtained from suppliers and used as received. See Appendix A for details. Water was purified using an ultrapure water system (Ulupure, Beijing, China) with a resistivity of 18.2 MΩ·cm and was used throughout the whole study.

**Standard procedure for the synthesis of Ir(IV)-doped PdAg NDs.** To prepare PdAg NDs, aqueous solutions of OTAC (200 mM, 1.25 mL), H_2_O (6 mL), Na_2_PdCl_4_ (20 mM, 500 μL), AgNO_3_ (20 mM, 250 μL), and AA (100 mM, 0.5 mL) were sequentially mixed and maintained at 60 °C in a water bath for 1 h. To prepare the Pd NDs, the procedure was maintained the same, except for the addition of AgNO_3_. To prepare the Ir-doped PdAg NDs, an aqueous solution of IrCl_3_ (10 mM, 200 μL) was directly added to one batch of the as-obtained PdAg NDs suspensions. This was maintained at 60 °C for another 1 h. The products were collected via centrifugation at 16,000 rpm for 10 min and washed with water once prior to further use and characterization.

**Instrumentations.** Transmission electron microscopy (TEM), high-resolution TEM (HRTEM), electron diffraction (ED), high-angle annular dark field-scanning transmission election microscopy (HAADF-STEM), and scanning transmission election microscope energy-dispersive X-ray spectroscopy (STEM-EDX) images were captured using a Talos F200X electron microscope (FEI, USA) operated at a 200 kV accelerating voltage. Scanning electron microscopy (SEM) images were obtained using a Zeiss Ultra60 microscope operated at 12 kV. The crystalline structures were recorded with a MiniFlex600 X-ray diffractometer (XRD, Rigaku, Tokyo, Japan). X-ray photoelectron spectroscopy (XPS) measurements were performed using a Thermo Fisher Scientific KALPHA XPS with monochromatic Al K_α_ radiation (*hν* = 1486.6 eV). Massive concentrations of metals were measured via inductively coupled plasma optical emission spectroscopy (ICP-OES, ICP-5000, FPI Group, China).

**Electrochemical measurements.** Electrochemical experiments of the formic acid oxidation reaction (FAOR) were carried out in a standard three-electrode system controlled by a CHI-760E potentiostat (CHInstruments, Shanghai, China). A glassy carbon electrode (3 mm in diameter), a Ag/AgCl electrode, and a platinum (Pt) wire were used as the working-, reference-, and counter-electrode, respectively. The potential was converted using Equation (1).
E (RHE) = E (Ag/AgCl) + 0.197 + 0.059 × pH(1)

Generally, the as-prepared Pd-based NDs were thoroughly washed with water several times to remove impurities and residual capping molecules. Then, 1 mg (in terms of Pd mass) of as-prepared Pd-based NDs and 2 mg of carbon black were dispersed in 1 mL of water and vigorously sonicated for 30 min. In total, 3 µL of the catalyst ink was drop-cast onto the glassy carbon electrode and dried naturally, followed by 3 µL of Nafion solution, which was also dried naturally. Prior to the FAOR measurements, electrocatalysts were first activated in nitrogen-saturated aqueous HClO_4_ solution (0.1 M) by CV cycling between 0.05 V and 1.05 V versus RHE at a scan rate of 200 mV s^−1^ until the reproducible curves were obtained. CV curves in aqueous HClO_4_ solution (0.1 M) with and without FA (0.5 M) were then collected sequentially between 0.05 V and 0.85 V versus RHE at a scan rate of 50 mV s^−1^. The chronoamperometric (CA) curves were recorded under a constant potential of 0.65 V vs. RHE. All electrochemical experiments were carried out at room temperature. The ECSA was estimated from the CV curve in aqueous HClO_4_ solution (0.1 M) using the following equation:$$\mathrm{ECSA}=\frac{Q_{\mathrm{PdO}}}{W\;\mathrm{mC}\cdot {\mathrm{cm}}^{-2}\times m}$$
where *Q_PdO_* denotes the charge, calculated by integrating the reduction peak area of PdO to Pd; *m* denotes the mass of Pd on the working electrode, as determined by ICP-OES; *W* is 0.424 mC·cm^−2^, which is the charge required for the reduction of the PdO monolayer.

**DFT calculation methods.** DFT simulations were based on a pseudo-potential plane wave method using the CASTEP program [38], and a cutoff energy of 400 eV, on the spin polarized GGA-PBE level [39]. In order to compare the catalytic ability, we built two surfaces including AgPd(111) and IrAgPd(111). On the basis of the Ag, Pd, Ir atom ratios, we replaced four surface Pd atoms with Ag atoms in the supercell (3 × 3) of Pd(111) to obtain AgPd(111). As for the ternary alloy of IrAgPd(111), we constructed the alloy of AgPd(111) by replacing one Ag atom with one Ir atom. As described in our recent work [40], the planes of the alloys were modeled with the supercell approach using three-layer slabs, in which the top layer was allowed to relax and the bottom layers were fixed. The vacuum layer was 10 Å above the surface to ensure the interaction between neighboring cells was negligible in the z direction. A Monkhorst–Pack grid of 2 × 2 × 1 was utilized for the integrations of the Brillouin zone.

## 4. Conclusions

In summary, we prepared Ir(IV)-doped PdAg nanodendrites with sub-5 nm branches and validated their promising use in the electrocatalytic FAOR. Ir doping was successfully realized by conducting an in situ galvanic replacement reaction, and the use of OTAC as a capping agent was crucial for the formation of the dendritic morphology. Compared to Pd NDs/C, PdAg NDs/C, and commercial Pt/C electrocatalysts, the Ir(IV)-doped PdAg NDs exhibited a large ECSA value and an enhanced electrochemical activity, together with improved reaction kinetics and long-term stability. DFT simulations showed that the Ir(IV)-doped PdAg NDs reduce the overall barrier in the oxidation of formic acid to CO_2_, shift the d-band center, and stabilize the diffusion of Pd atoms, boosting catalytic performance during the FAOR. The current work opens a new avenue for engineering Pd-based trimetallic nanocrystals with versatile control over the morphology and composition, shedding light on the design of advanced fuel cell electrocatalysts.

## Figures and Tables

**Figure 1 molecules-28-03670-f001:**
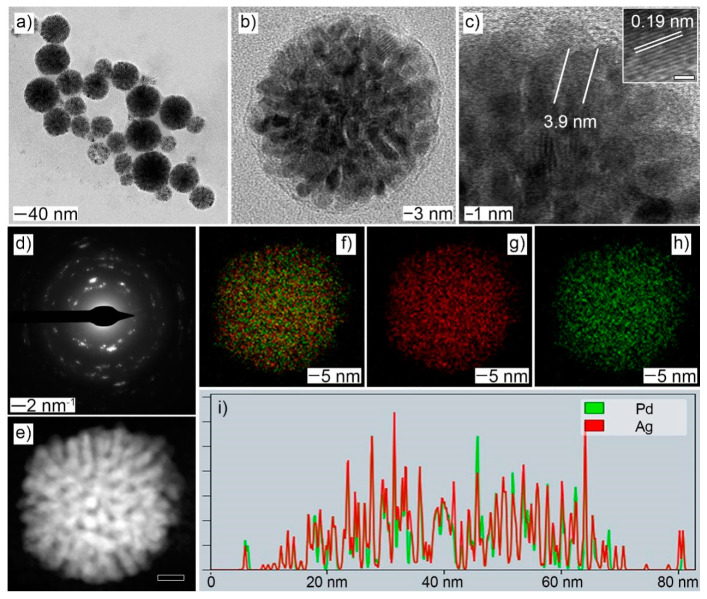
Morphology and structural characterizations of PdAg NDs: (**a**,**b**) TEM; (**c**) HRTEM, inset: lattice fringe with the scale bar of 1 nm; (**d**) SAED; (**e**) HAADF-STEM, scale bar: 10 nm; (**f**–**h**) EDX-STEM mapping: (**f**) Pd + Ag; (**g**) Ag; (**h**) Pd; (**i**) line-scan profile, X-axis: distance and Y-axis: Intensity (a. u.).

**Figure 2 molecules-28-03670-f002:**
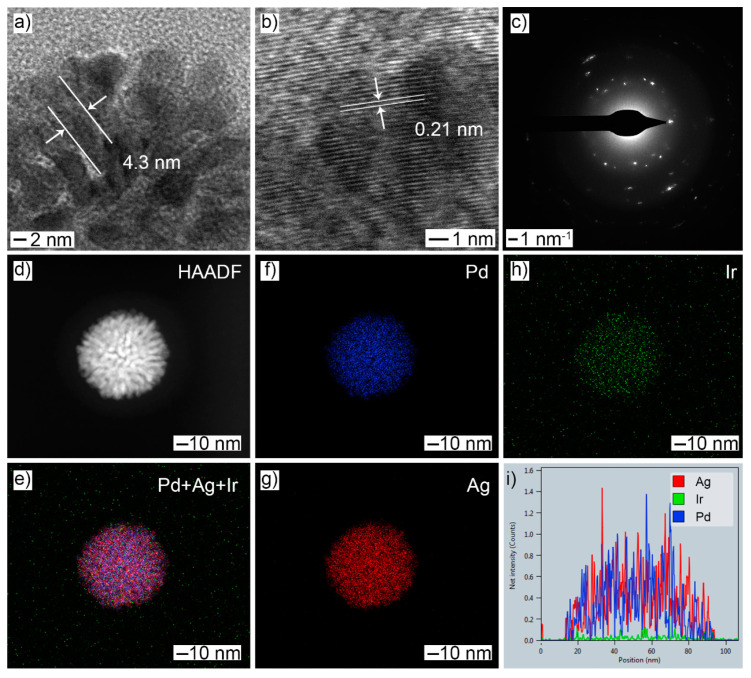
Morphology and structural characterizations of PdAg NDs after the GRR treatment: (**a**) TEM; (**b**) HRTEM; (**c**) SAED; (**d**) HAADF-STEM; (**e**–**h**) EDX-STEM mapping: (**e**) Pd + Ag + Ir; (**f**) Pd; (**g**) Ag; (**h**) Ir; (**i**) line-scan profile.

**Figure 3 molecules-28-03670-f003:**
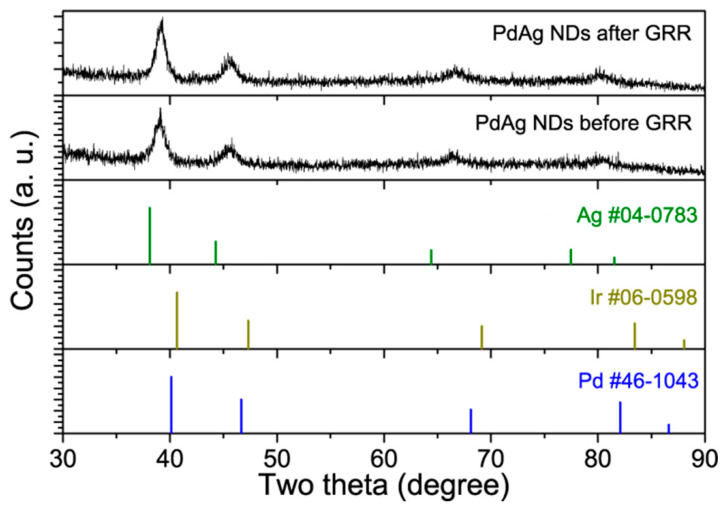
XRD patterns of PdAg NDs before and after the GRR treatment. JCPDS data of *fcc*-Ag, *fcc*-Ir, and *fcc*-Pd were plotted for reference.

**Figure 4 molecules-28-03670-f004:**
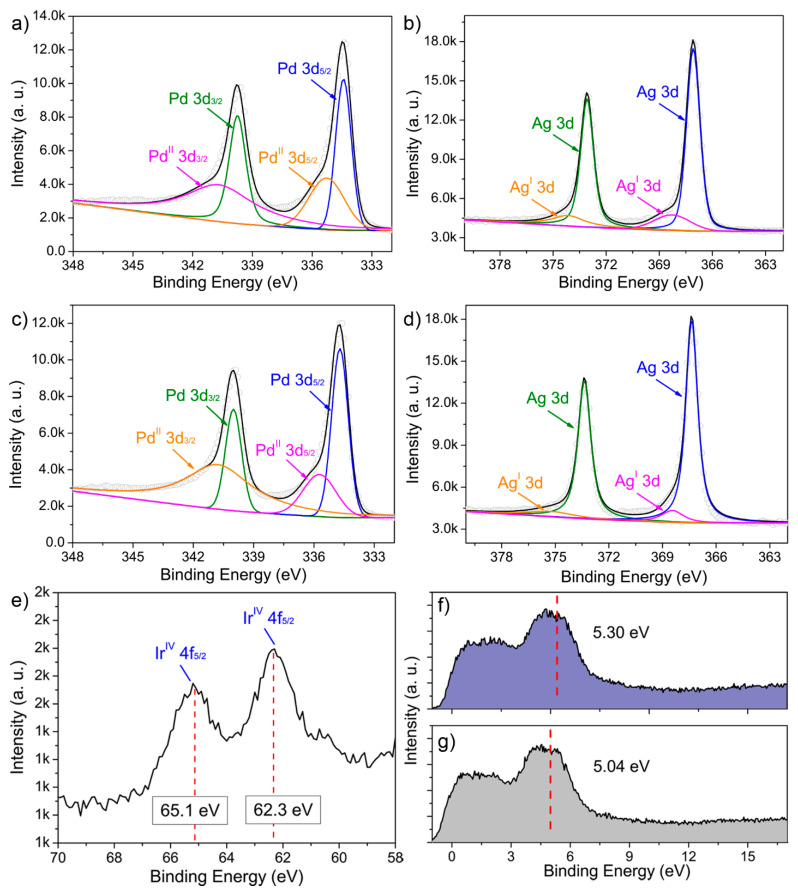
XPS analyses. (**a**,**c**) Pd 3d and (**b**,**d**) Ag 3d high-resolution XPS spectra of PdAg NDs (**a**,**b**) before and (**c**,**d**) after the GRR treatment. (**e**) Ir 4f high-resolution XPS spectrum of GRR-treated PdAg NDs. (**f**,**g**) VB-XPS spectra of PdAg NDs (**f**) after and (**g**) before the GRR treatment. The red line indicated the d-band center.

**Figure 5 molecules-28-03670-f005:**
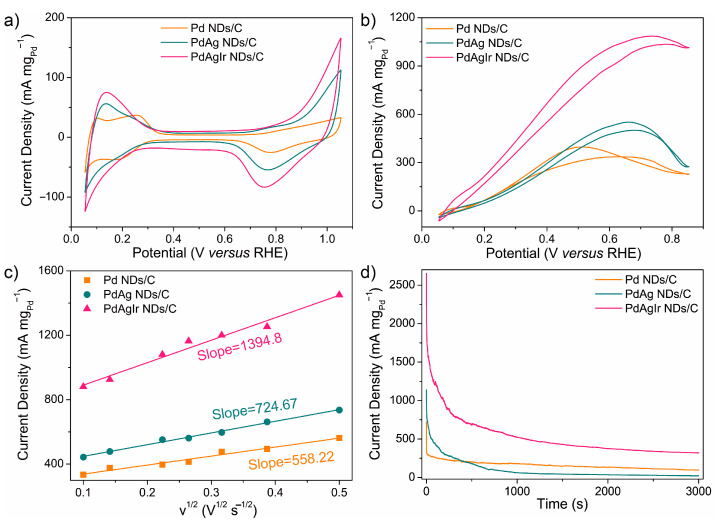
FAOR measurements of PdAg NDs/C and PdAgIr NDs/C: (**a**) CV traces in 0.1 M HClO_4_; (**b**) CV traces in 0.1 M HClO_4_ + 0.5 M FA; (**c**) plots showing FAOR kinetics versus scan rates; (**d**) chronoamperometric (CA) stability curves measured at 0.65 V versus RHE.

**Figure 6 molecules-28-03670-f006:**
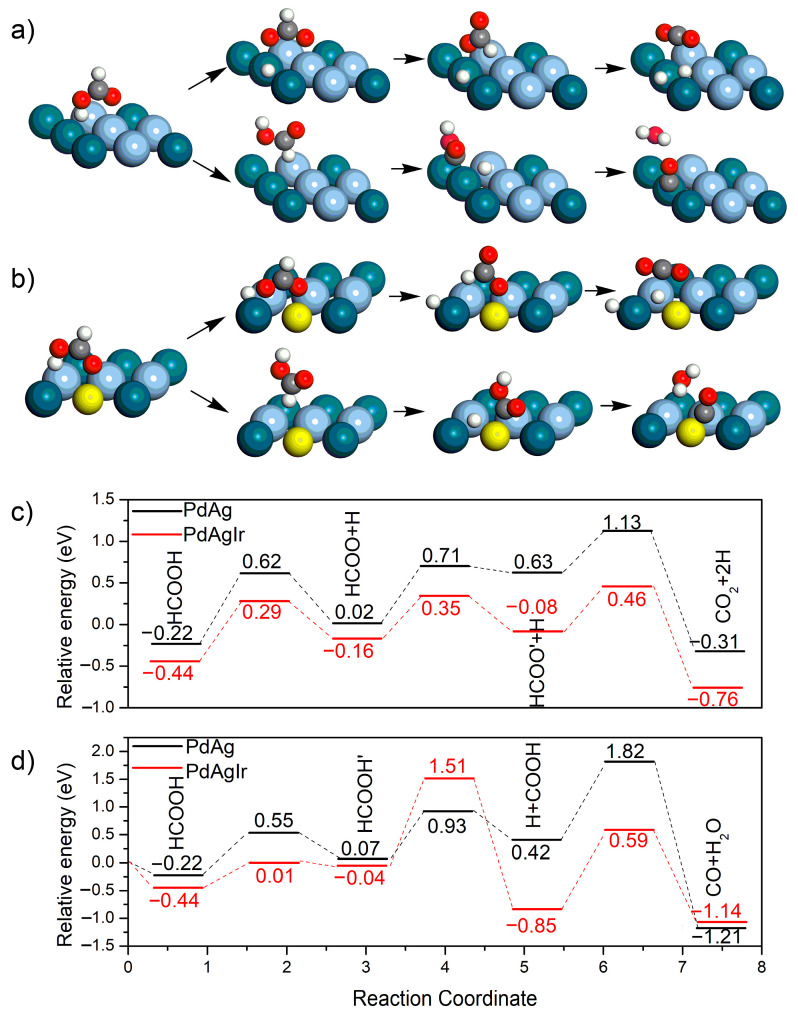
Theoretical simulations. (**a**,**b**) Schematic representation of possible reaction pathways of FAOR on (**a**) AgPd(111) and (**b**) IrAgPd(111). (i) and (ii) indicate the direct oxidation pathway and the CO poisoning pathway, respectively. For clarity, only the top layer of the slab is shown. Dark blue, light blue, yellow, red, gray, and white spheres represent Pd, Ag, Ir, O, C, and H atoms, respectively. (**c**) The relative energy profiles of the direct oxidation pathways. (**d**) The relative energy profiles of the CO poisoning pathway.

**Figure 7 molecules-28-03670-f007:**
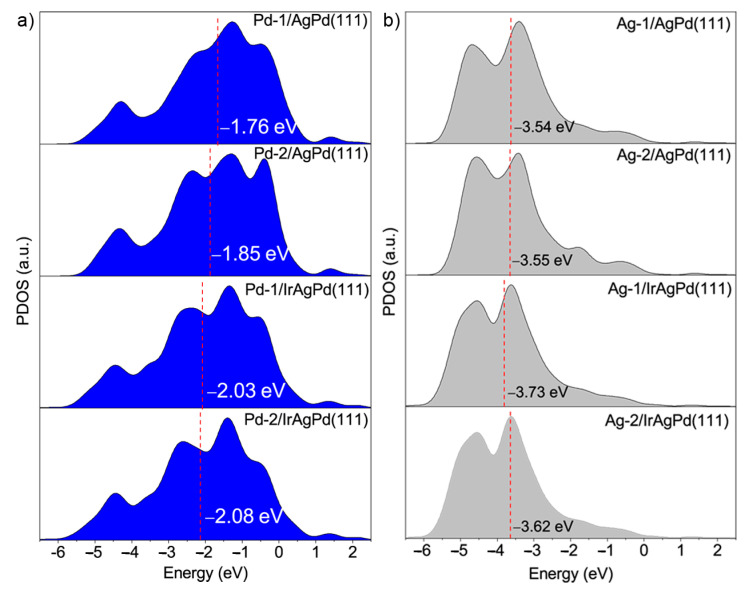
Projected d-band PDOS of (**a**) Pd and (**b**) Ag atoms for AgPd(111) and IrAgPd(111) model systems. The d-band center is marked by a red line. The Fermi level was determined to be 0 eV.

**Figure 8 molecules-28-03670-f008:**
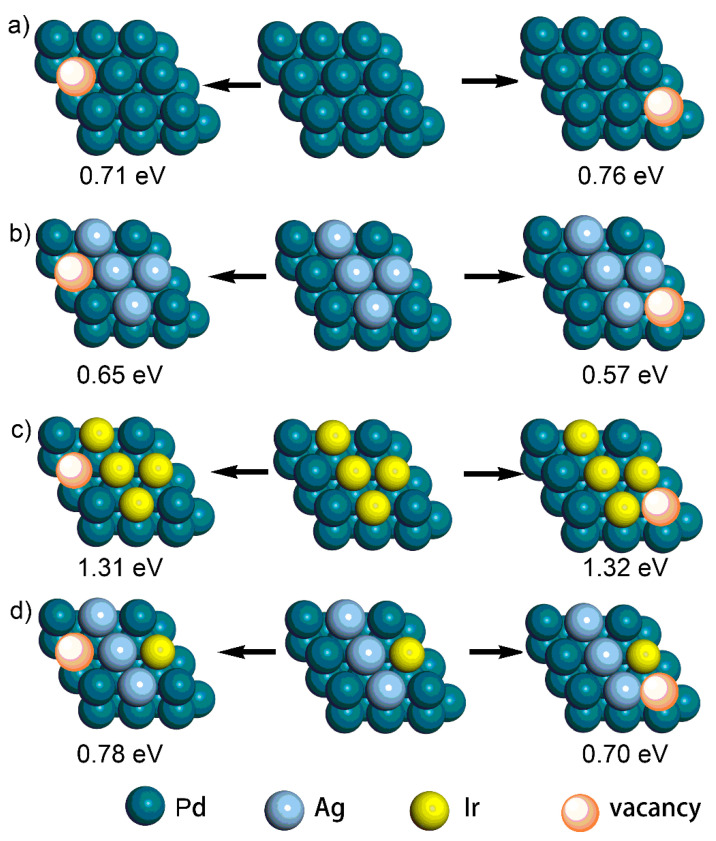
The calculated results of the vacancy energy (E_v_) of Pd atoms on the (111) facets (top view) of (**a**) Pd, (**b**) AgPd, (**c**) IrPd, and (**d**) IrAgPd.

## Data Availability

Data available on request from the authors.

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
