# Peer review of "Boosting Electrocatalytic Oxidation of Formic Acid on Ir(IV)-Doped PdAg Alloy Nanodendrites with Sub-5 nm Branches"

_molecules, 2023, doi:10.3390/molecules28093670_

Round 1
Reviewer 1 Report
Recommendation: This paper represents a significant new contribution and should be published as is.Comments:
The authors have studied the fabrication of Ir-doped PdAg alloy nanodendrites with sub-5 nm branches via a stepwise synthesis, where the precursors of Pd and Ag are co-reduced, followed by the addition of IrCl3 to conduct in situ galvanic replacement reaction.
When they are serving as the electrocatalysts for FAOR in acidic medium, the Ir doping unambiguously enhances the activity of PdAg alloy nanodendrites, as well as improved reaction kinetics and long-term stability. In particular, the carbon-supported PdAgIr nanodendrites exhibits a prominent mass activity with a value of 1.09 A mgPd-1, which is almost 2.0-times and 2.7-times that of PdAg and Pd counterparts, and far superior to that of the commercial Pt/C. As confirmed by means of DFT simulations, such improved electrocatalytic performance stems from the reduced overall barrier in the oxidation of formic acid into CO2 during FAOR and successful d-band tuning, together with the stabilization of Pd atoms. The current study opens a new avenue for engineering Pd-based trimetallic nanocrystals with versatile control over the morphology and composition, shedding light on the rational design of advanced fuel cell electrocatalysts.
The calculation procedure and the level of theory are excellent and the calculated results are reasonable and well support the authors’ conclusion. The manuscript is well organized and written. Therefore, I recommend this manuscript for publication as it is.
Author Response
Thanks for your valuable comment. Your suggestion/comment on the present study is highly appreciated.
Reviewer 2 Report
In this study, the authors reported the Ir-doped PdAg alloy nanodendrites with sub-5 nm branches, where the Pd and Ag precursors were initially co-reduced, followed by the addition of Ir via in situ galvanic replacement reaction. The obtained catalysts were investigated for FAOR catalysis and DFT simulations were conducted to explain the mechanism for enhancement in electrocatalytic activity. This referee would like to recommend the acceptance of current work after addressing the following issues.
1. The authors highlighted the function of Ir doping for FAOR. To this end, some of recent progresses of Ir-based electrocatalysts should be discussed in the introduction section.
2. Workup process of electrocatalysts should be described in the experimental section.
3. The quality of some figures should be improved. For example, the font in the last figure is not uniform.
4. Recent advances in rational design of electrocatalysts should be cited to enrich the background, such as 10.1002/bkcs.12588; 10.1021/acscatal.1c05766; 10.1073/pnas.2112109119.
Author Response
In this study, the authors reported the Ir-doped PdAg alloy nanodendrites with sub-5 nm branches, where the Pd and Ag precursors were initially co-reduced, followed by the addition of Ir via in situ galvanic replacement reaction. The obtained catalysts were investigated for FAOR catalysis and DFT simulations were conducted to explain the mechanism for enhancement in electrocatalytic activity. This referee would like to recommend the acceptance of current work after addressing the following issues.
- The authors highlighted the function of Ir doping for FAOR. To this end, some of recent progresses of Ir-based electrocatalysts should be discussed in the introduction section.
Thanks for this valuable comment. We have added the discussion in Line 57-68. In particular, among recent studies, we have witnessed the rapid development in rational design of high-performance electrocatalysts based on the Ir-doping strategy. For example, Guo and coworkers reported report an iridium-tungsten alloy with a nanodendritic structure as a new class of high-performance and pH-universal bifunctional electrocatalysts for hydrogen and oxygen evolution catalysis, which presents a hydrogen generation rate ∼2 times higher than that of the commercial Pt/C catalyst in both acid and alkaline media. Yu and coworkers developed a facile approach to control the composition of IrPt alloy nanoparticles without affecting the size using an antisolvent crystallization-based method owing to the separation of the particle growth and precursor reduction steps, which exhibited satisfactory electrocatalytic performances in both hydrogen evolution reaction and oxygen reduction reaction. Henkelman and coworkers report a rational synthetic strategy toward the preparation of sub-10 nm Rh-Ir nanoparticles as highly efficient oxygen evolution reaction catalysts under acidic conditions based on microwave-assisted synthesis. All these successful demonstrations have illustrated the feasibility to involve Ir as a promising catalytic element in constructing efficient electrocatalysts. Related studies have also been cited (see Ref. 31-33).
- Workup process of electrocatalysts should be described in the experimental section.
Thanks for this valuable comment. Generally, the as-prepared Pd-based NDs were thoroughly washed with water for several times to removal impurity and residual capping molecules. We have added this content in Line 275-276.
- The quality of some figures should be improved. For example, the font in the last figure is not uniform.
Thanks for this valuable comment. We have improved the qualities of Figure 6 and 8 by unifying the fonts.
- Recent advances in rational design of electrocatalysts should be cited to enrich the background, such as 10.1002/bkcs.12588; 10.1021/acscatal.1c05766; 10.1073/pnas.2112109119.
Thanks for this valuable comment. W e have cited the studies suggested by the reviewer in the main text to enrich the background of current manuscript (see Ref. 7, 30, and 37).
Reviewer 3 Report
The authors of the manuscript have prepared Ir-doped Pd-Ag alloy nanodendrites via a stepwise synthesis for the electrocatalytic oxidation of formic acid. Overall, this work is relevant and interesting, and I would like to recommend the acceptance by the Molecules Journal after addressing the following comments.
1) In lines 136-138, the authors suggested that Iridium species were present on the surface of the Pd-Ag particles. Moreover, there are no peaks of zero-valent Iridium on the XPS spectra. Therefore, the authors are not dealing with a ternary alloy but rather a metal system of Pd-Ag/IrO2. In that case, the oxygen line on the EDX-STEM mapping should be visible. How can the authors explain this?
2) Also, I recommend identifying the SAED pattern in Fig. 1d and Fig. 2c to be more persuasive in responding to the first comment.
Author Response
The authors of the manuscript have prepared Ir-doped Pd-Ag alloy nanodendrites via a stepwise synthesis for the electrocatalytic oxidation of formic acid. Overall, this work is relevant and interesting, and I would like to recommend the acceptance by the Molecules Journal after addressing the following comments.
1) In lines 136-138, the authors suggested that Iridium species were present on the surface of the Pd-Ag particles. Moreover, there are no peaks of zero-valent Iridium on the XPS spectra. Therefore, the authors are not dealing with a ternary alloy but rather a metal system of Pd-Ag/IrO2. In that case, the oxygen line on the EDX-STEM mapping should be visible. How can the authors explain this?
Thanks for this valuable comment. We agree with the reviewer that based on the XPS result, the Ir species should be present in a high oxidation state, possibly IrO2. To this end, we conducted extra EDS measurement towards the current sample and the results have been included in Figure S3. We note that in addition to the peaks attributed to the three metallic elements, O Kα line located at 0.525 keV could be identified, suggesting the possible formation of IrO2 after the galvanic replacement reaction. Corresponding discussion have been added in line 128-131.
However, we would like to propose the concern that the presence of IrO2 phase should be demonstrated with more solid evidences, such as XRD. However, due to the limited proportion of Ir-based species in the current products, we did not observe noticeable diffraction peaks in the XRD pattern that can be clearly indexed to the IrO2 phase. Meanwhile, the oxygen signal from EDS measurement may also be possibly contributed by the partial oxidation of Pd, as revealed by XPS results in Fig. 4c. Taken together, based on the current data, to provide a precise description of the composition of the current product, we have replaced the “Ir-doped” with “Ir(IV)-doped” in the title and revised the main text and figures accordingly for clarification.
2) Also, I recommend identifying the SAED pattern in Fig. 1d and Fig. 2c to be more persuasive in responding to the first comment.
Thanks for this valuable comment. We agree with the reviewer that the IrO2 phase should have been reflected in the SAED pattern. However, due to its limited amount and the intrinsic polycrystalline structure of current Ir(IV)-doped PdAg nanodendrites, the SAED pattern had become too complicated to be accurately labeled. To this end, we regretfully acknowledge that we are unable to precisely isolate and identify the exact set of diffraction spots belonging to IrO2 from the SAED patterns shown in Fig. 1d and Fig. 2c and expect more advanced techniques could be developed to facilitate this characterization.